# Effects of Hypocalcemic Vitamin D Analogs in the Expression of DNA Damage Induced in Minilungs from hESCs: Implications for Lung Fibrosis

**DOI:** 10.3390/ijms23094921

**Published:** 2022-04-28

**Authors:** Esmeralda Magro-Lopez, Irene Chamorro-Herrero, Alberto Zambrano

**Affiliations:** Chronic Diseases Program, Institute of Health Carlos III, 28220 Madrid, Spain; esme-1988@live.com (E.M.-L.); irenechamorrrohrro@gmail.com (I.C.-H.)

**Keywords:** human pluripotent stem cells, hESCs, minilungs, vitamin D, vitamin D analogs, paricalcitol, calcipotriol, lung fibrosis

## Abstract

In our previous work, we evaluated the therapeutic effects of 1α,25-Dihydroxyvitamin D_3_, the biologically active form of vitamin D, in the context of bleomycin-induced lung fibrosis. Contrary to the expected, vitamin D supplementation increased the DNA damage expression and cellular senescence in alveolar epithelial type II cells and aggravated the overall lung pathology induced in mice by bleomycin. These effects were probably due to an alteration in the cellular DNA double-strand breaks’ repair capability. In the present work, we have evaluated the effects of two hypocalcemic vitamin D analogs (calcipotriol and paricalcitol) in the expression of DNA damage in the context of minilungs derived from human embryonic stem cells and in the cell line A549.

## 1. Introduction

DNA damage and cellular senescence underlie the physiopathology associated with idiopathic pulmonary fibrosis (IPF) and other chronic conditions that can evolve with fibrosis. IPF is a form of progressive interstitial pneumonia of unknown etiology with an estimated survival of 3 to 4 years [1]. IPF pathogenesis is the consequence of an excessive matrix deposition leading to tissue scarring and irreversible organ injury, probably due to a persistent input of damage and tissue repair response. It has been reported that cellular senescence is implicated in the tissue repair program, and its occurrence in IPF, unfortunately, has a detrimental role in contrast to other fibrogenic conditions [2,3,4]. Vitamin D and its analogs have been demonstrated to be active in the regulation of fibrosis that characterizes multiple chronic diseases, including pulmonary fibrosis [5,6,7,8]. For instance, the preventive use of vitamin D supplementation was associated with a general improvement in the lung fibrosis symptomatology induced in mice, probably due to its anti-inflammatory effects [8,9]. However, as we have reported in our previous work, the “therapeutic” treatment of mice having bleomycin-induced fibrosis seemed to worsen the pathology: the mice treated with vitamin D showed increased architectural distortion, subpleural scarring and more areas of aberrant reepithelization compared to the controls. These areas were defined by the accumulation of alveolar epithelial type II (ATII) cells harboring high levels of DNA damage in the form of DNA double-strand breaks (DSBs). DSBs were also observed in cells throughout respiratory bronchioles or immersed in alveolar fields. The bulk of the DNA damage was preferably associated with epithelial cells; fibroblasts, however, seemed to be more resistant to DNA damage than epithelial cells [10]. Senescence can be induced prematurely as a result of a persistent DNA damage response (DDR) secondary to oxidative stress that induces DNA double-strand breaks [11]. Indeed, DSBs are potent inducers of cell arrest and a typical hallmark of cell senescence [12]. Our results also showed significantly greater levels of DSBs and cell senescence in epithelial cells than in fibroblasts, and they were consistent with the central hypothesis underlying IPF indicating that epithelial injury and impaired regeneration activate fibroblasts and that cellular senescence induced by persistent epithelial damage may be the origin of aberrant epithelial regeneration and the promotion of fibrosis [13,14,15,16].

Although bleomycin reproduces well many aspects of the general pulmonary fibrosis and some lesions present in IPF, it has never been promoted as an experimental equivalent of IPF. The strength of the bleomycin model consists of the reproducibility and versatility as a model of general fibrosis. In addition, its high efficiency levels inducing DSBs make bleomycin a very interesting model to analyze DNA damage [17].

In the present study, we have evaluated various vitamin D analogs in the context of DNA damage induced by bleomycin. A large amount of vitamin D analogs have been synthesized throughout the years, and the clinical use for secondary hyperparathyroidism, osteoporosis or psoriasis has been approved for many of them [18,19]. The potent effect of vitamin D on intestinal calcium and phosphorus absorption and bone mineral mobilization, often leading to the development of hypercalcemia and hyperphosphatemia, has precluded its therapeutic use for many conditions. The ideal analog would retain vitamin D receptor-binding capacities and have minimal effects on calcium and phosphorus metabolism. Our working hypothesis is that hypocalcemic vitamin D analogs could show a lower incidence in the expression of DNA damage upon a bleomycin insult than the active form of vitamin D.

## 2. Results 

We have tested our postulated hypothesis in the cell line A549, an immortalized counterpart of ATII cells, in 2D minilungs (lung and alveolar differentiated cells from hESCs arranged in bidimensional cultures) and in 3D minilungs from hESCs (human lung bud organoids embedded in Matrigel^TM^ sandwiches). The lung organoids generated from hESCs have enormous advantages over cell lines or simple primary cultures as they offer an unlimited availability of primary cells, show the complete lung epithelial spectrum and emulate structural and functional features of the original organ.

The exposure of A549 cells to a sublethal bleomycin shock (12 μg/mL for 6 h) induces the expression of DNA damage (DD) foci containing TP53BP1, a reliable marker of DSBs [20,21,22,23,24]. These conditions allow the accurate quantification of DSBs. To avoid a potential effect of bleomycin on the beginning of the transcription triggered by vitamin D or its analogs, we have performed short pre-treatments with these ligands before the addition of bleomycin. In addition, as vitamin D and its analogs show differential binding affinities to their receptor, we assessed increasing amounts of ligands in the dose–response assay using the expression of the gene *CYP24A1*, a vitamin D target gene, as the response. The results showed that, in the case of A549 cells or 2D minilungs, the amount of 5 nmol/L is adequate as we found no significant differences between vitamin D and the analogs evaluated in terms of *CYP24A1* expression. However, in the case of the minilungs embedded in Matrigel^TM^ sandwiches (3D organoids), this low concentration of ligand gave rise to significant differences between vitamin D and the analogs evaluated, probably due to the limited access of the ligands to the organoid cells. The amount of 50 nmol/L, however, resulted in similar responses of vitamin D and the hypocalcemic analogs (Appendix A).

After bleomycin treatments, DD foci can be rapidly visualized as discrete foci in a pan-nuclear pattern (Figure 1A). As previously reported [10], the exposure of A549 cells to vitamin D, in the presence of bleomycin, increased the levels of DD foci, both the percentage of damaged cells and the levels of severely damaged cells harboring more than 20 DD foci per nucleus (Figure 1B–D) (*n* = 3; >150 cells were analyzed; *p* < 0.001). However, the two hypocalcemic vitamin D analogs tested (paricalcitol and calcipotriol) were able to drastically reduce the bulk of the DD expression compared to vitamin D in the presence of bleomycin (Figure 2C,D; *n* = 3; >150 cells were analyzed; ANOVA *p* < 0.001). Figure 2A,B shows that the treatment of A549 cells with vitamin D or its analogs, in the absence of bleomycin, did not alter the low basal level of damage of the cell population as previously described for vitamin D [10] (*n* = 3; >150 cells were analyzed; ANOVA *p* < 0.001). In order to reproduce these results in much more reliable models of lung structure and function, we generated minilungs from hESCs as previously described [25,26]. On the one hand, we generated lung airway and epithelial cells arranged in bidimensional cultures (2D minilungs) from the hESC line AND-2 as previously described [25]. Briefly, good hESCs colonies are grown along inactivated MEFs (iMEFs), picked up and passaged to new plates with iMEFs in order to accumulate material for lung differentiation. Figure 3A shows the expression of pluripotency marker SOX-2 in a good AND-2 colony and representative micrographs at various times of the differentiation process: embryoid bodies (EBs), anterior foregut endoderm (AFE), cultures at day 23 (lung progenitors) and at day 60 (differentiated lung airway and alveolar cells). The cultures from day 50 onwards show the expression of representative markers of the lung airway and alveolar cells, illustrating the heterogeneity in cell shape, including the presence of flat cells with a crescent shape morphology and granular and roughly cuboidal-shaped cells, likely corresponding to ATI and ATII cells, respectively (Figure 3A; d60). Although from day 50 they can be considered mature, the cultures were used for the desired experimentation from day 60 on. Figure 3B shows a RT-qPCR result illustrating the complexity of these cultures (*n* = 3; > 4 organoids per experimental replicate were used; ANOVA *p* < 0.001). As previously described by us and others [25,26,27,28,29], the differentiation protocol applied here yields cultures enriched in alveolar epithelial cells (ATI and ATII cells). On the other hand, the generation of 3D minilungs implies the formation of nascent organoids in suspension at certain times of the protocol (see Figure 3C and Methods for details) and their final embedding in MatrigelTM sandwiches to reach the desirable state of differentiation characterized by the presence of lung buds more or less branched [lung buds organoids (LBOs)], as previously described [26,29] (Figure 3D). Figure 3E shows representative micrographs of histochemical analysis (H&E staining) and immunohistochemical analysis with surfactant antibodies performed on LBOs sections. In order to analyze the expression of DNA damage and the effect of vitamin D and its analogs, 2D minilungs were treated with 12.5 μg/mL of bleomycin for 72 h. All the cell types of these complex cultures seemed to be affected equally by the bleomycin treatment. As in the case of the A549 cells, neither vitamin D nor its hypocalcemic analogs altered significantly the basal levels of DD in the absence of bleomycin (Figure 4A,B) (*n* = 3; >150 cells were analyzed; ANOVA *p* < 0.001). As expected, the exposure of these cultures to bleomycin and vitamin D increased the levels of DD foci reached by bleomycin itself (Figure 4C,D). As in the case of the A549 cells, the treatment with paricalcitol and calcipotriol did not further increase the DD levels reached by bleomycin and seemed to reduce significantly the DD expression induced by bleomycin (Figure 4C,D, (*n* = 3; >4 organoids per condition were used and >150 cells were analyzed; ANOVA *p* < 0.001). Equivalent assays were performed on 3D minilungs embedded into matrigelTM sandwiches. As previously reported, lung buds minilungs are mainly constituted by ATII cells [26,29]. Although to a lesser extent, we found similar results to those obtained in the case of 2D minilungs (Figure 4E–H) [*n* = 3; >4 organoids per condition were used and >150 cells were analyzed; ANOVA *p* > 0.05 (panels 4E,F) and ANOVA *p* < 0.001 for data represented in panels 4G,H]. This reduction in the extension of damage was probably because the organoids are embedded into matrigel^TM^ sandwiches and the access of bleomycin and the ligands to them is more restricted. Finally, we evaluated, in A549 cells, a continuous cell line counterpart of ATII cells, the expression of γH2AFX marker, which is a reliable marker of DD, cell senescence and aging, as previously described [20,30]. Figure 4I shows the significant increase in the γH2AFX expression levels in the presence of bleomycin and vitamin D compared to the controls and the drastic reduction induced by paricalcitol. Equivalent assays were performed using a battery of commercially available less-hypercalcemic vitamin D analogs, including 22-oxacalcitriol, tacalcitol and vitamin D2. All the vitamin D analogs seemed not to further increase the expression levels of γH2AFX reached in bleomycin-treated cells (Figure 4J). Moreover, in the case of paricalcitol, a significant reduction in the expression of γH2AFX compared to the bleomycin-treated cells was observed.

## 3. Discussion

Besides its function in the mineral homeostasis and immune system, vitamin D plays a role in multiple chronic diseases involving the respiratory system. Epidemiological studies have suggested a link between vitamin D deficiency and the risk of development of chronic lung diseases, such as asthma, chronic obstructive pulmonary disease (COPD), cystic fibrosis and respiratory infections [31]. This association has led to the notion that vitamin D supplementation might ameliorate the progress of these diseases. Vitamin D supplementation, however, needs to be evaluated carefully as it can be a factor contributing to vitamin-D-mediated hypercalcemia and hypercalciuria [32]. In addition, cases of vitamin D toxicity associated with overdoses due to manufacturing or intake errors have been reported [33]. Moreover, we have reported a detrimental role of vitamin D supplementation in a therapeutic experimental system, very likely associated with an impairment in the cellular DSBs repair capabilities and cell senescence [10]. 

Vitamin D may affect the progression of fibrosis at different stages: anti-fibrinolytic coagulation cascade, inflammation, fibroblasts activation and on the negative regulation of the renin–angiotensin system. Vitamin D seemed to prevent the experimental lung fibrosis induced by bleomycin [8,9,34,35,36]. However, in these experimental studies, vitamin D is administered either before or very early after the bleomycin insult, so the effects observed were very likely due to the inherent anti-inflammatory properties of vitamin D. Thus, these studies can be defined as preventive. In addition, various hypocalcemic analogs, such as paricalcitol, calcipotriol and 22-oxacalcitriol, have been demonstrated to be active as anti-fibrotic agents in different experimental systems and types of fibrosis [37,38,39,40,41,42,43,44,45,46,47,48,49]. Vitamin D less-hypercalcemic analogs might provide an alternative to vitamin D supplementation to treat many conditions related to fibrosis. The ideal analog would retain vitamin D receptor-binding capacities and have minimal effects on mineral metabolism. 

Bleomycin can induce pulmonary fibrosis and fibrogenic cytokine release by oxidant-mediated DNA scission in a variety of animal models. The principal drawbacks of the bleomycin model relate to the rapid lung remodeling and the emphysema-like changes induced [17]. However, it reproduces well many aspects of the general pulmonary fibrosis and some lesions present in IPF, although it should be stated here that bleomycin is not a reliable model of IPF [17]. However, the potential of bleomycin in the induction of DSBs and senescence in many cell types is extraordinary [10,13]. We have made use of this advantage in the present study to explore the influence of various vitamin D hypocalcemic analogs in the context of A549 cells and minilungs generated from hESCs. Our experiment approaches the initial steps of the fibrogenic conditions, i.e., the expression of DNA damage underlying many conditions evolving towards fibrosis. 

The generation of human minilungs that share the structural features and some extent of the functionality of the native organ may serve as a system model to emulate the DNA damage inflicted during the course of fibrogenic conditions, such as IPF. Currently, the more efficient protocols to generate airway and alveolar epithelial cells from the direct differentiation of hPSCs are biased to the production of alveolar cells [26,27,29]. We have employed either shocks or continuous exposures of bleomycin. The sublethal bleomycin shocks employed here allow the accurate quantification of DNA damage in the form of DSBs and the observation of subtle differences between the experimental conditions that might otherwise be masked by the extraordinary potential of bleomycin. Bleomycin seems to inflict DNA damage in the form of DSBs in all the epithelial cells equally, even when the cell organization is the form of lung buds embedded in Matrigel^TM^ sandwiches. However, the assembly of organoids into Matrigel^TM^ sandwiches can make difficult the access of bleomycin and ligands to the cells. The reduction in the extent of DNA damage inflicted by bleomycin compared to the 2D minilungs or A549 cells might reflect this fact. 

As a continuation of our earlier work [10], our current hypothesis states that hypocalcemic vitamin D analogs could show a lower incidence in the expression of DNA damage upon a bleomycin insult than the active form of vitamin D. The initial results presented here suggest that less-hypercalcemic analogs do not show the deleterious effects observed by vitamin D treatment in the presence of bleomycin and could be an alternative to vitamin D supplementation. In addition, the treatment with such vitamin D analogs could be tested as efficient agents to reduce the bulk of the DD expression underlying multiple diseases that can evolve with DNA damage, fibrosis and aging, such as IPF and other lung interstitial conditions. Future in vivo work in this direction will be necessary.

## 4. Materials and Methods

### 4.1. Cell Culture

Alveolar epithelial cells type II (A549, ATCC) were maintained in DMEM medium supplemented with 10% FBS (Sigma-Aldrich, Burlington, MA, USA), 2 mM glutamine and 100 U/mL of penicillin and streptomycin (Lonza, Basel, Switzerland). We used the active form of vitamin D (1α,25-Dihydroxyvitamin D_3_ or calcitriol) (cat.#D1530; Sigma-Aldrich; Vitamin D stock was 10 μM in ethanol) and the following vitamin D analogs (stocks were 50 μM in ethanol): calcipotriol (cat.#203537; Santa Cruz Biotechnology, Dallas, TX, USA), paricalcitol (cat.#477938; Santa Cruz Biotechnology), tacalcitol (cat.#sc-361371a; Santa Cruz Biotechnology, Dallas, TX, USA), 22-Oxacalcitriol (cat.#sc-361076; Santa Cruz Biotechnology, Dallas, TX, USA) and vitamin D2 (cat.#sc- sc-205988; Santa Cruz Biotechnology, Dallas, TX, USA). Treatments were performed in cells maintained in DMEM supplemented with 10% hormone-depleted serum. This serum was prepared by using the anion exchange resin AGR1-X8 from BIO-RAD (cat.#1401441; BIO-RAD, Hercules, CA, USA) as previously described [20]. Bleomycin sulfate (cat.# CAYM13877–50) was purchased to VWR [Radnor, PA, USA] (bleomycin stock: 50 mM in PBS). 

### 4.2. Maintenance of hESCs

The hESCs line AND-2 was obtained from the “Biobanco de células madre de Granada” (ISCIII, Spain); passages 26–40. Mouse embryonic fibroblasts (MEFs) were obtained at 13.5 days post-coitum from C57BL/6 mice as described previously [20]. MEFs were mitotically inactivated by an overnight treatment with 2 µg/mL of mitomycin C (cat.#M4287; Sigma-Aldrich, Darmstatdt, Germany) and plated at a density of approximately 16,000 cells/cm^2^. hESCs were cultured along with MEFs under standard conditions (http://www.stembook.org (accessed on 5 April 2022)). The maintenance medium was composed of KO-DMEM (cat.#10829-018 Gibco; Life Technologies, Carlsbad, CA, USA), 20% KO serum replacement (cat.#10828010 Gibco; Life Technologies, Carlsbad, CA, USA), 0.1 mM β-mercaptoethanol (cat.#21985-023 Gibco; Life Technologies, Carlsbad, CA, USA), 2 mM Glutamax (cat.#35050-061,Gibco; Life Technologies, Carlsbad, CA, USA), nonessential aminoacids (cat.#11140-050 Gibco; Life Technologies (Carlsbad, CA, USA) and primocin (cat.#12I05MM; InvivoGen, Toulouse, France). The medium was filtered by using 0.22-µ pore filter systems (cat.#431097; Corning, NY, USA); 10 ng/mL recombinant human basic fibroblast growth factor (hbFGF) (cat.#PHG6015; Invitrogen, Waltham, MA, USA) and 10 µM Y-27632 (cat.#1254; Tocris, R&D Systems, Bristol UK) were added before use. The medium was changed on a daily basis and cells were passaged either by enzymatic (collagenase IV method) (collagenase IV: cat.#11140050; Gibco; Life Technologies, Carlsbad, CA, USA) or mechanical procedures (http://stembook.org (accessed on 5 April 2022)). Cells were maintained in an undifferentiated state in a 5% CO_2_/air environment. The differentiation process was carried out under hypoxic conditions in a 5% CO_2_/5% O_2_/95% N_2_ environment (Galaxy 48R incubator (New Brunswick); Eppendorf, Hamburg, Germany) or in normoxy, as indicated in the corresponding differentiation step. 

### 4.3. Primitive Streak Formation and Induction of Definitive Endoderm (DE)

Induction of endoderm was performed as previously described [25,26]. Primitive streak formation (day 0; 24 h) and endoderm induction (days 1–4) were performed in serum-free differentiation (SFD) medium. SFD medium was composed of a mix of IMDM:F12 (3:1) media (cats.#B12-722F and 10-080 CVR; Corning, Corning, NY, USA), supplemented with N2 (cat.#17502-048, Gibco; Life Technologies, Carlsbad, CA, USA), B27 (cat.#17504-044, Gibco; Life Technologies, Carlsbad, CA, USA), 2 mM Glutamax (cat.#35050-061 Gibco; Life Technologies, Carlsbad, CA, USA), 1% penicillin-streptomycin (DE17-602E; Lonza, Basel, Switzerland) and 0.05% bovine serum albumin (BSA) (cat.#A7906; Sigma-Aldrich, Burlington, MA, USA). The medium was filtered using a 0.22 µ-pore filter system (cat.#431097; Corning, Corning, NY, USA); 50 μg/mL ascorbic acid (cat.#A4554; Sigma-Aldrich, Burlington, MA, USA) and 0.04 μL/mL monothioglycerol (stock > 97%) (cat.#M6145; Sigma-Aldrich, Burlington, MA, USA) were added before use. MEFs were depleted by passaging hESCs lines onto Matrigel^TM^-coated (cat.#354230; Life Technologies, Carlsbad, CA, USA) plates for at least 48 h. Cells were briefly trypsinized into small 3–10 cell clumps and the reaction was halted with stop medium (IMDM medium (BE12722F) supplemented with 50% foetal bovine serum (F7524, Sigma-Aldrich, Burlington, MA, USA), 2 mM Glutamax, 1% penicillin-streptomycin and 30 ng/mL DNase I (cat.#260913-10MU; Calbiochem, San Diego, CA, USA). Cells were then centrifuged 5 min at 850 rpm and washed carefully two times with an excess of SFD medium. To form embryoid bodies (EBs), the clumps were plated onto low-attachment 6-well plates (cat.#3471; Corning, Corning, NY, USA) and maintained in SFD medium in a 5% CO_2_/5% O2/95% N_2_ environment (Galaxy 48R incubator; New Brunswick). 

For primitive streak formation, 10 μM Y-27632, 10 ng/mL Wnt3a (cat.#5036-WN; R&D Systems, Minneapolis, MN, USA) and 3 ng/mL human BMP4 (cat.#314-BP; R&D Systems, Minneapolis, MN, USA) were used. EBs were collected, resuspended carefully in endoderm induction medium containing 10 μM Y-27632, 0.5 ng/mL human BMP4, 2.5 ng/mL hbFGF and 100 ng/mL human Activin (cat.# 338-AC; R&D Systems, Minneapolis, MN, USA). Cells were fed after 36–48 h, depending on cell density, by removing half the old medium and adding half fresh medium.

### 4.4. Induction of Anterior Foregut Endoderm (AFE) 

AFE (days 4, 5 or 5) was induced as previously described [25,26]. EBs were dissociated into single cells with trypsin. Dissociated cells were transferred to a conical tube containing stop medium to neutralize trypsin. Cells were centrifuged for 5 min at 850 rpm, washed carefully twice with SFD medium and counted. For AFE induction, 25,000–30,000 cells/cm^2^ were plated on fibronectin-coated (F0895; Sigma-Aldrich, Burlington, MA, USA, USA) 12-well tissue culture plates in AFE induction medium 1 (SFD medium supplemented with 10 mM SB-431542 (cat.#1614; Tocris, Bristol, UK) and 100 ng/mL of NOGGIN (cat.#6057; R&D Systems, Minneapolis, MN, USA). After 24 h of incubation, the medium was aspirated and AFE induction medium 2 (SFD medium supplemented with 1 µM IWP2 (cat.#3533; Tocris, Bristol, UK) and 10 µM of SB-431542) was added to the cultures. This process was carried out under hypoxic conditions only for the bidimensional cultures. 

### 4.5. Lung Progenitors Induction and Expansion 

Lung progenitor induction and expansion was carried out as previously described [25,26]. On day 6,5–7, AFE cultures treated for 20 days with the ventralization medium consisting of SFD medium supplemented with 3 µM CHIR99021 (cat.#04; Tocris, Bristol, UK), 10 ng/mL human FGF10 (cat.#345-FG; R&D Systems, Minneapolis, MN, USA), 10 ng/mL human KGF (cat.#251KG-010; R&D Systems, Minneapolis, MN, USA), 10 ng/mL human BMP4 (cat.#314-BP; R&D Systems, Minneapolis, MN, USA), 10 ng/mL murine EGF (cat.#2028-EG-200; R&D Systems, Minneapolis, MN, USA) and 50 nM all-trans retinoic acid (cat.#R2625; Sigma-Aldrich, Burlington, MA, USA). The culture medium was changed every two days. At a time point between days 8 and 12, cultures were incubated under normoxic conditions. At day 16, cultures were briefly digested with trypsin in order to remove potential nonectodermal contaminating cells. Supernatant of this brief digestion containing single cells and small clumps were removed. The remaining cell clumps were replated onto fibronectin-coated MW12 plates at 1:3 dilutions in fresh medium after trypsin neutralization and careful washing. Plates were returned to the hypoxic conditions (5% CO_2_/5%O_2_/95%N_2_ environment).

### 4.6. Lung and Airway Epithelial Cells Maturation 

At day 26, cultures were incubated with SFD medium supplemented with 3 µM CHIR99021, 10 ng/mL human FGF10, 10 ng/mL human FGF10, 0.1 mM 8-bromocAMP (cat.# B5386; Sigma-Aldrich, Burlington, MA, USA), 0.1 mM IBMX (3,7-dihydro-1-methyl-3-(2methylpropyl)-1H-purine-2,6-dione; cat.# I5879; Sigma-Aldrich, Burlington, MA, USA) and 60 nM dexamethasone (cat.#D5902; Sigma-Aldrich, Burlington, MA, USA). The medium was changed every two days and plates were maintained under hypoxic conditions (5%CO_2_/5%O_2_/95%N_2_ environment). Cultures were carried further under these conditions until their experimental use at day 50. Treatments were performed in minilungs maintained in day 26 medium as indicated in the corresponding experiments. 

### 4.7. Formation of Lung Bud Organoids 

In this case, the differentiation process was performed under normoxic conditions from the anteriorization stage on. At day 8, cells were briefly trypsinized into small 3–10 cell clumps and the reaction was halted with stop medium (IMDM medium (BE12-722F) supplemented with 50% fetal bovine serum (FBS; F7524; Sigma-Aldrich, Burlington, MA, USA), 2 mM Glutamax, 1% penicillin-streptomycin). Cells were then centrifuged for 5 min at 850 rpm and washed carefully twice with an excess of SFD medium. The clumps were plated onto low-attachment six-well plates (cat.#3471; Corning, Corning, NY, USA) in branching medium (SFD medium containing 3 μM CHIR99021, 10 ng/mL FGF10, 10 ng/mL KGF, 10 ng/mL BMP4, 50 nM all-trans retinoic acid). These three-dimensional clumps (nascent lung bud organoids) were incubated and fed every other day for approximately 20–25 days. After that, these nascent organoids were embedded into a Matrigel^TM^ sandwich assembled on MW96 wells. 50 μL of Matrigel^TM^ were loaded on the MW96 well and allowed to gel. Nascent organoids were picked up with a wide mouth plastic Pasteur pipette, divided into MW96 wells containing 50% Matrigel^TM^, diluted in branching media and immediately transferred onto the first layer of Matrigel^TM^. After solidification of this intermediate layer containing the nascent organoids, 50 μL of Matrigel^TM^ were added on top. Finally, each sandwich containing various organoids was incubated with 50 μL branching media. Medium was changed every 2–3 days. Growing branching structures were easily visualized under the microscope after 1 or 2 weeks. Treatments were performed in minilungs maintained in branching medium as indicated in the corresponding experiments. 

### 4.8. Indirect Immunofluorescence of A549 Cells and 2D Minilungs

Cells were seeded in 8-well chambers (cat.#154,534; Thermofisher Scientific, Waltham, MA, USA) at a density of 20,000 cells/well. The following day, the cells were treated as indicated in the corresponding experiments. Immunofluorescence was performed as previously described [20]. Basically, the cells were fixed in 2% PFA in PBS for 10 min at RT and permeabilized with 0.1% Triton X-100 and 0.1% sodium citrate for 5 min at RT. Preparations were washed with PBS and washing solution (PBS/0.25% BSA/0.1% Tween 20), blocked for 30 min with blocking solution (washing solution + 2.5% BSA) and incubated overnight with antibodies against TP53BP1 (1:500; sc-16565; Invitrogen, Waltham, MA, USA). Preparations were then washed with washing solution and incubated with secondary antibodies conjugated with Alexa fluor dyes (488, 546) from Life Technologies (cat.#A-11029, cat.#A-11035) for 1 h at RT. Nuclei were counterstained with DAPI, and samples were mounted with ProLong Diamond (cat.#P36961; Life Technologies, Waltham, MA, USA). Cell images were captured with fluorescence microscope (Zeiss Axio) equipped with a camera (AxiocamMRm) and AxioVision software. DNA damage foci were quantified by counting from >150 cells for each experimental condition. For 2D minilungs, the glass chamber slides were incubated overnight at 4 ºC with human fibronectin in order to plate the differentiated cells. Cultures from day 50 were digested with trypsin, neutralized with stop medium and washed with SFD medium. Approximately 40,000 differentiated epithelial cells per well were plated in the epithelial maturation medium. Cultures were maintained under normoxic conditions for one day before treatments. 

### 4.9. Indirect Immunofluorescence of Lung Bud Organoids 

Organoids were picked up from the MW96 wells, transferred into a well of a MW12 and fixed with 4% paraformaldehyde (PFA) for 15 min at RT. After that, the organoids were washed three times with PBS for 10 min and incubated overnight at 4 °C with 30% sucrose. The sucrose was exchanged for a solution of 7.5% gelatin/15% sucrose and incubated for 15 min at 37 °C. The organoids were carefully transferred to cryomolds and progressively embedded in various layers of solidified 7.5% gelatin/15% sucrose. These preparations were cut into 10-μm sections in a Leica CM3050 cryostat. The mounted sections were washed with PBS and permeabilized with PBS/1% BSA/0.25% Triton X-100 for 5 min at RT. After that, the sections were washed and blocked for 30 min at RT with blocking solution (PBS-BSA 1%). The sections were incubated for 2 h with antibodies against TP53BP1 (1:500; sc-16565; Invitrogen, Waltham, MA, USA) or the pro surfactant protein C (1:200; ab3785, Merck). Preparations were washed with washing solution and incubated with a secondary antibody conjugated with Alexa fluor dye (546) from Life Technologies (cat.#A-11035, Waltham, MA, USA) for 1 h at room temperature. Nuclei were counterstained with DAPI and samples were mounted with ProLong Diamond (cat.#P36961; Life Technologies, Waltham, MA, USA). Cell images were captured with a fluorescence microscope (Zeiss Axio) equipped with a camera (AxiocamMRm) and AxioVision software. DNA damage foci were counted from >150 cells for each experimental condition.

### 4.10. Analysis of Proteins by Western Blot 

Cell monolayers were washed with ice-cold PBS and lysed in triple-detergent lysis buffer (50 mM Tris-HCl pH 8.0, 150 mM NaCl, 0.02% sodium azide, 0.1% SDS, 1% NP-40, 0.5% sodium deoxycholate, 100 μg/mL PMSF, 2 μg/mL pepstatin, 2 μg/mL aprotinin, 2 μg/mL leupeptin, and phosphatase inhibitors cocktail 2 or 3 (cat.#P5726, P0044, Sigma-Aldrich, St. Louis, MO, USA)). SDS-PAGE and immunoblotting were performed under standard conditions. Basically, samples in Laemmli buffer (30 μg/lane) were separated through 12% gels and transferred to nitrocellulose membranes for 90 min at RT in the presence of 20% methanol and 0.1% SDS. Membranes were blocked with 3% BSA in PBS-Tween 0.05% (PBST-BSA) and incubated O/N at 4 °C with a γH2AFX antibody (cat.#05–636, Millipore, Burlington, MA, USA) diluted 1:1000 in PBST-BSA. Densitometry analysis of bands was performed by using Image J software (https://imageJ.nih.gov/ (accessed on 22 April 2020)).

### 4.11. Quantitative real-time RT-PCR (RT-qPCR) of minilungs

Total RNA was extracted using Trizol (cat.#15596026; Ambion) following manufacturer’s instructions. cDNA was generated using the High-Capacity cDNA kit (cat.#4387406; Applied Biosystems, Waltham, MA, USA). Real-time qPCR was performed by using the powerUpSYBR Green mix (cat.#A25742) on the Quantstudio-3 system (Applied Biosystems, Waltham, MA, USA) following manufacturer’s instructions. Absolute quantification of each gene was obtained using a standard curve of serial diluted genomic DNA (cat.#11807720, Roche) and normalized to housekeeping gene TBP (TATA box binding protein).

The genes analyzed and the sequences of the oligonucleotides employed in this study were the following: TBP [Tata-Box Binding Protein; Forward: 5′-TGAGTTGCTCATACCGTGCTGCTA, Reverse: 5′-CCCTCAAACCAACTTGTCAACAGC]; TP63 (Tumor Protein P63, marker of basal cells) [Forward: 5′-CCTATAACACAGACCACGCGCAGA, Reverse: 5′-GTGATGGAGAGAGAGCATCGAAG]; MUCIN5AC (Mucin 5AC, marker of globet cells) [Forward: 5′GCACCAACGACAGGAAGGATGAG, Reverse: 5′-CACGTTCCAGAGCCGGACAT]; SCGB1A1 (Secretoglobin Family1A Member1 or CC10, marker of clara cells) [Forward: 5′-TCATGGACACACCCTCCAGTTATGAG.

Reverse: 5′-TGAGCTTAATGATGCTTTCTCTGGGC]; *PDPN* (Podoplanin, marker of AT-I cells) [Forward: 5′- AGGAGAGCAACAACTCAACGGGA, Reverse: 5′- TTCTGCCAGGACCCAGAGC]; *AQP5* (Aquaporin 5, marker of AT-I cells) [Forward: 5′- GCCATCCTTTACTTCTACCTGCTC, Reverse: 5′- GCTCATACGTGCCTTTGATGATGG]; *SFTPA* (Surfactant Protein A, marker of AT-II cells) [Forward: 5′-GTGCGAAGTGAAGGACGTTTGTG, Reverse: 5′-TTTGAGACCATCTCTCCCGTCCC]; *SFTPB* (Surfactant Protein B, marker of AT-II cells) [Forward: 5′-TCTGAGTGCCACCTCTGCATGT, Reverse: 5′-TGGAGCATTGCCTGTGGTATGG]; *SFTPC* (Surfactant Protein C, marker of AT-II cells) [Forward: 5′-CCTTCTTATCGTGGTGGTGGTGGT, Reverse: 5′-TCTCCGTGTGTTTCTGGCTCATGT]; *SFTPD* (Surfactant Protein D, marker of AT-II cells) [Forward: 5′-TGACTGATTCCAAGACAGAGGGCA, Reverse: 5′-TCCACAAGCCCTGTCATTCCACTT]; *FOXJ1* (Forkhead Box J1, marker of ciliated cells) [Forward: 5′-GGCATAAGCGCAAACAGCCG, Reverse: 5′-TCGAAGATGGCCTCCCAGTCAAA]; *CYP24A1* (cytochrome P450 family 24 subfamily A member 1) [Forward: 5′-GGTGACATCTACGGCGTAC, Reverse: 5′-CTTGAGACCCCCTTTCCAGAG]. 

### 4.12. Statistical Analysis 

Data were subjected to the Shapiro–Wilk test and D’Agostino and Pearson omnibus test to verify their normality. Statistical significance of data was determined by applying a two-tailed Student’s *t*-test or analysis of variance followed by the Newman–Keuls or Bonferroni post hoc tests for experiments with more than two experimental groups; *p* < 0.05 is considered significant. Significance of analysis of variance post hoc test or the Student’s *t*-test is indicated in the figures as *, *p* < 0.05; **, *p* < 0.01 and ***, *p* < 0.001. Statistics were calculated with the Prism 9 software (GraphPad Software). The results presented in the figures are means ±SEM. Experiments were repeated three times.

## 5. Conclusions

The bleomycin treatment of cells and organoids might be used to reproduce the bulk of the DNA damage expression underlying multiple conditions that evolve with fibrosis. The treatment with vitamin D less-hypercalcemic analogs does not increase the DNA damage expression reached by the sole treatment with bleomycin. In terms of DNA damage expression, these initial results indicate that the vitamin D analog paricalcitol behaves as a potential DNA damage eraser. This fact might be exploited to set up a therapy for fibrogenic diseases. 

## Figures and Tables

**Figure 1 ijms-23-04921-f001:**
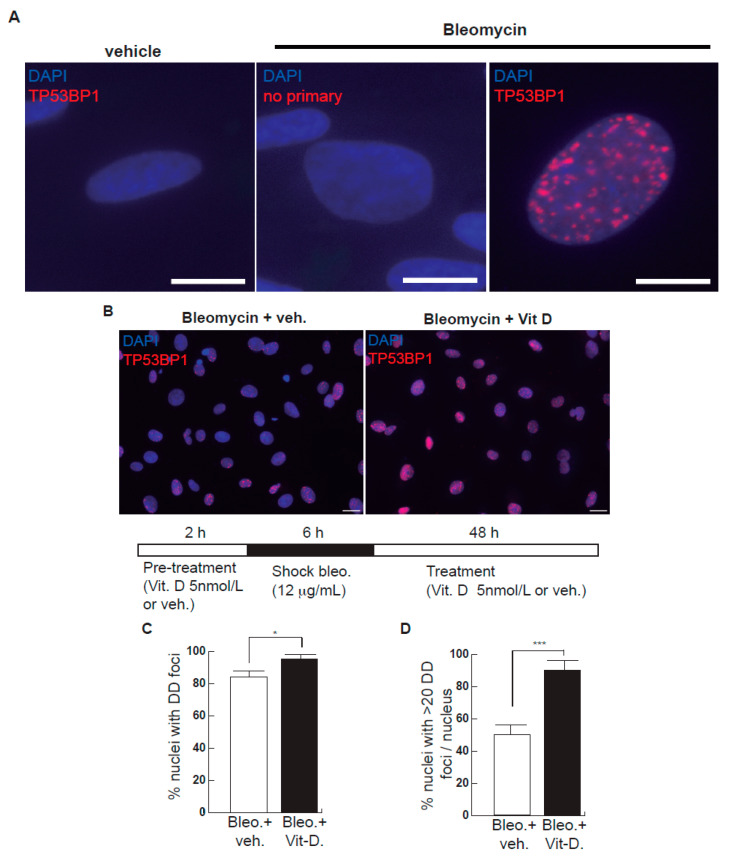
Effects of bleomycin and vitamin D in A549 cells. (**A**) A549 cells were treated with bleomycin with a sublethal bleomycin shock (12 μg/mL for 6 h) incubated for 48 h and then processed for indirect immunofluorescence to detect DNA damage foci containing TP53BP1 (red dots), a reliable marker of DNA double-strand breaks (DSBs). Veh.: bleomycin vehicle (PBS). No primary: negative control of the immunofluorescence assay consisting of the absence of primary antibody. Scale bar: 5 μm. (**B**) Expression of DSBs (TP53BP1 foci (red dots)) induced by vitamin D in the presence of bleomycin. The cells were pre-treated with 5 nmol/L vitamin D for 2 h, subjected to a bleomycin shock (12 μg/mL for 6 h) and then treated with 5 nmol/L vitamin D or its vehicle. Representative micrographs taken at 48 h post-shock are shown. Scale bar: 20 μm. (**C**,**D**) Quantification of damaged cell: % of nuclei with TP53BP1 foci (**C**) or % of severely damaged cells (more than 20 TP53BP1 foci per nucleus (**D**)). Data from three experiments are represented; more than 150 cells per condition were analyzed (*p* < 0.001). *: *p* < 0.05, ***: *p* < 0.001.

**Figure 2 ijms-23-04921-f002:**
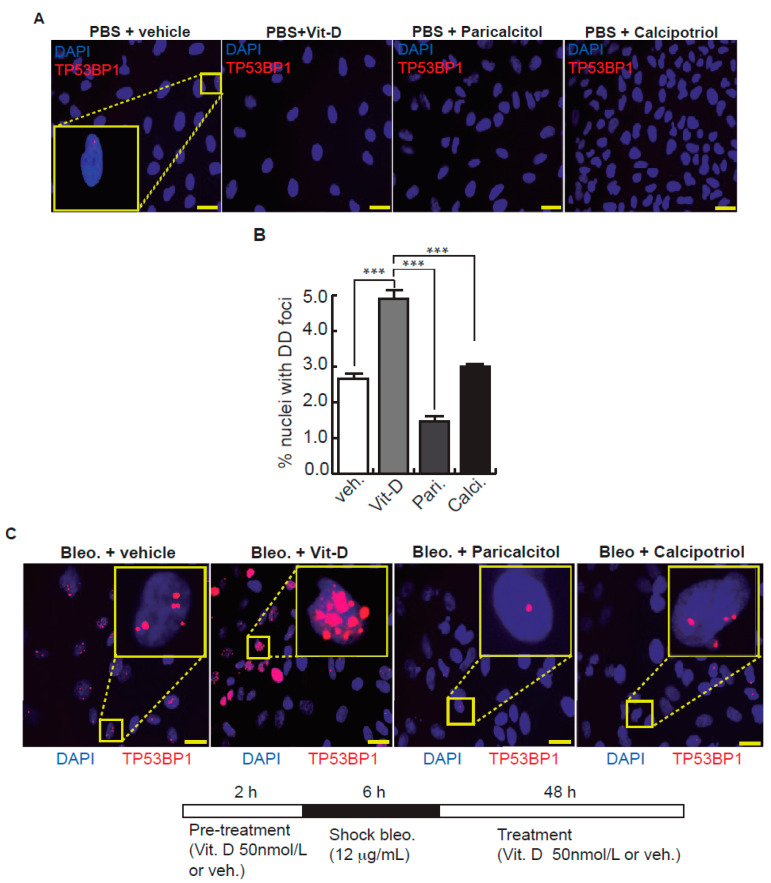
Effects of vitamin D and two hypocalcemic analogs in the expression of DNA damage induced by bleomycin in A549 cells. (**A**) Basal expression of DSBs (TP53BP1 foci (red dots)) induced by vitamin D and the analogs paricalcitol and calcipotriol in the absence of bleomycin. A549 cells were treated with 50 nmol/L vitamin D (or the corresponding analog) for 48 h and then subjected to immunofluorescence to detect DD foci. Representative micrographs are shown. PBS is the bleomycin vehicle; vehicle: vitamin D or analogs vehicle; scale bar: 10 μm. (**B**) Quantification of damaged cells (nuclei with TP53BP1 foci). Data from three experiments are represented; more than 150 cells per condition were analyzed. ANOVA *p* < 0.001. (**C**) Expression of TP53BP1 foci (red dots) in cultures of A549 cells pre-treated with 50 nmol/L vitamin D, analogs or its vehicle for 2 h and subjected to a bleomycin shock (12 μg/mL) for 6 h. After that, the cultures were treated with 50 nmol/L vitamin D and the analogs paricalcitol and calcipotriol (or vehicle) for 48 h. Representative micrographs taken at 48 h post-shock are shown. Scale bar: 10 μm. (**D**) Quantification of damaged cells; veh.: vitamin D or analogs vehicle. Data from three experiments are represented; more than 150 cells per condition were analyzed; ANOVA *p* < 0.001. The results presented in the figures are means ± SEM. Significance of the analysis is indicated as *: *p* < 0.05, **: *p* < 0.01, ***: *p* < 0.001.

**Figure 3 ijms-23-04921-f003:**
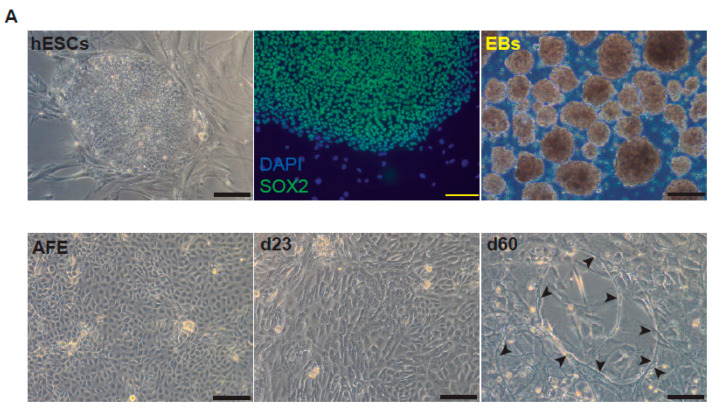
Representative micrographs of the sequential differentiation processes and expression markers. (**A**) Left upper micrograph: AND-2 colony growing along with feeder cells (inactivated MEFs (iMEFs)); scale bar: 100 μm); central upper micrograph: expression of SOX2 (SRY (sex-determining region Y)-box 2) in an undifferentiated colony of AND-2; scale bar: 100 μm. Right upper micrograph: representative micrograph of EBs (embryoid bodies). Bottom panels: AFE (anterior foregut endoderm) and representative micrographs of cultures at day 23 of differentiation (lung progenitors) and at day 60 (differentiated lung and airway cells); black arrowheads signal cells with a typical flat and crescent shape morphology denoting alveolar type I cells (ATI cells); scale bar: 100 μm. (**B**) Levels of expression (relative to TBP (TATA box binding protein)) of lung and airway epithelial cells markers at day 60 (*n* = 3; > 4 organoids per condition were used). (**C**) Representative micrograph of nascent organoids growing in suspension at day 23. (**D**) Representative micrograph of LBOs at day 50 embedded in Matrigel^TM^ sandwiches; scale bar: 100 μm and 50 μm (micrograph on the right). (**E**) Micrograph on the left: histochemical analysis of LBO sections (H&E staining); representative micrograph of an immunohistochemical staining of LBO sections (micrograph on the right) with an SFTP-C antibody. Scale bar: 100 μm.

**Figure 4 ijms-23-04921-f004:**
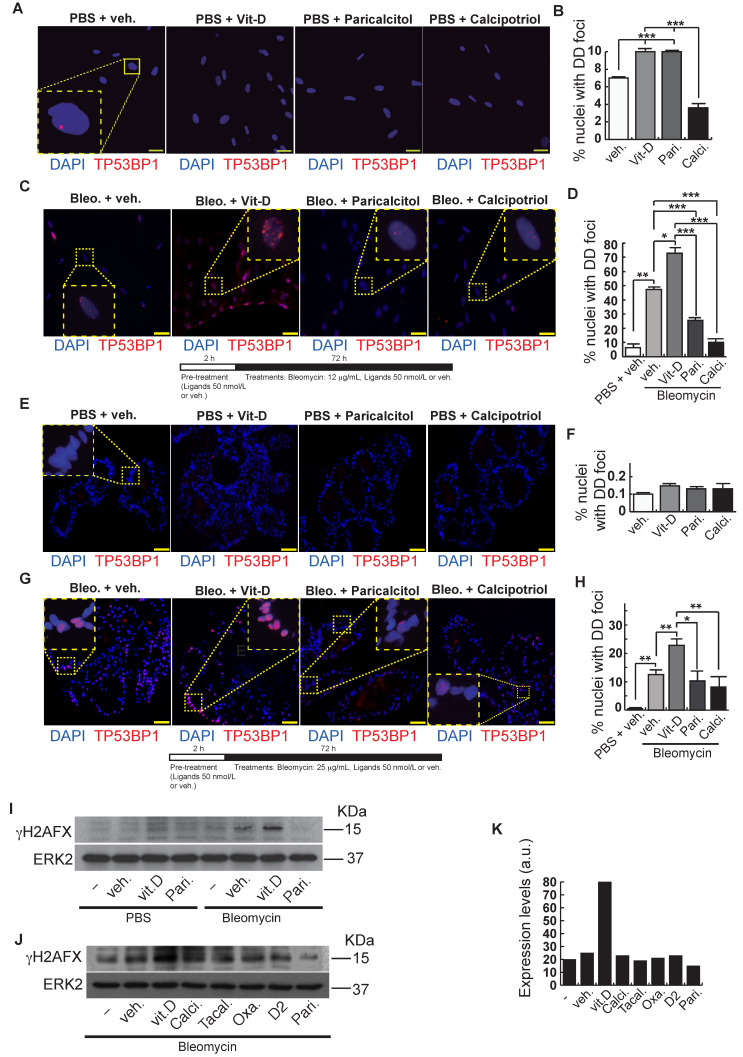
Effects of vitamin D and hypocalcemic analogs in the expression of DNA damage induced by bleomycin in the minilungs generated from hESCs. (**A**) Basal expression of DNA damage (TP53BP1 foci (red dots)) induced by vitamin D and the analogs paricalcitol and calcipotriol in the absence of bleomycin. 2D minilungs were treated with 50 nmol/L vitamin D (or the corresponding analog) for 72 h and then subjected to immunofluorescence to detect DD foci. Representative micrographs are shown. PBS is the bleomycin vehicle; veh.: vitamin D or analogs vehicle; scale bar: 10 μm. (**B**) Quantification of damaged cells (nuclei harboring TP53BP1 foci) corresponding to conditions A; (*n* = 3; >150 cells were analyzed; ANOVA *p* < 0.001). (**C**) Expression of TP53BP1 foci (red dots) in 2D minilungs cells pre-treated with 50 nmol/L vitamin D, analogs or its vehicle for 2 h and then with bleomycin (12.5 μg/mL), vitamin D or analogs (50 nmol/L) for 72 h. After that, the cultures were processed for indirect immunofluorescence to detect DNA damage foci containing TP53BP1 (red dots); representative micrographs are shown. Scale bar: 10 μm. (**D**) Quantification of damaged cells corresponding to conditions C. veh.: vitamin D or analogs vehicle; (*n* = 3; >150 cells were analyzed; ANOVA *p* < 0.001). (**E**) Basal expression of DNA damage in 3D minilungs (LBOs): TP53BP1 foci (red dots) induced by vitamin D and the analogs paricalcitol and calcipotriol in the absence of bleomycin. 3D minilungs (LBOs) were treated with 50 nmol/L vitamin D (or the corresponding analog) for 72 h and then subjected to immunofluorescence to detect DD foci. Representative micrographs are shown. PBS is the bleomycin vehicle; veh.: vitamin D or analogs vehicle; scale bar: 10 μm. (**F**) Quantification of damaged cells corresponding to condition E (nuclei harboring TP53BP1 foci); (*n* = 3; >4 organoids per condition were used and >150 cells were analyzed; ANOVA *p* > 0.05). (**G**) Expression of TP53BP1 foci (red dots) in 3D minilungs (LBOs) cells pre-treated with 50 nmol/L vitamin D, analogs or its vehicle for 2 h and then with bleomycin (25 μg/mL), vitamin D or analogs (50 nmol/L) for 72 h. After that, the cultures were processed for indirect immunofluorescence to detect DNA damage foci containing TP53BP1 (red dots); representative micrographs are shown. Scale bar: 10 μm. (**H**) Quantification of damaged cells corresponding to condition G (nuclei harboring TP53BP1 foci); (*n* = 3; >4 organoids per condition were used and >150 cells were analyzed; ANOVA *p* < 0.001). The results presented in the figures are means ± SEM. Significance of the analysis is indicated as *: *p* < 0.05, **: *p* < 0.01, ***: *p* < 0.001. (**I**) Detection of γH2AFX in A549 cell extracts. Cells were pre-treated with 50 nmol/L vitamin D for 2 h, subjected to a bleomycin shock (12 μg/mL for 6 h) and then treated with 50 nmol/L vitamin D or its vehicle. Cell extracts were obtained at 48 h post-shock; Pari: paricalcitol; ERK2 was used as loading control. KDa: kilodaltons. (**J**) Detection of γH2AFX in A549 cell extracts. Cells were treated as in J. Cell extracts were obtained at 48 h post-shock; Calci: calcipotriol; Tacal: tacalcitol; Oxa.: 22-oxacalcitriol; D2: vitamin D2; Pari: paricalcitol; ERK2 was used as loading control. KDa: kilodaltons. (**K**): densitometry analysis of western J; a.u: arbitrary units.

## Data Availability

Please contact the corresponding author for data requests.

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
