# Peer review of "Effects of Hypocalcemic Vitamin D Analogs in the Expression of DNA Damage Induced in Minilungs from hESCs: Implications for Lung Fibrosis"

_ijms, 2022, doi:10.3390/ijms23094921_

Round 1

Reviewer 1 Report

I think that the data presented is interesting, however I am not entirely sure of the correctness of the model. Although the bleomycin model is commonly used in mice because it can reproduce many aspects of IPF, administration of bleomycin is more suitable for inducing acute lung injury. For this reason I think that bleomycin treatment may be useful in inducing DNA damage but cannot be considered an absolute model of IPF. I believe that this observation should at least be reflected in the limitations of the study, which are not present.

Furthermore, the discussion is extremely heated. I advise the authors to cite and comment on other articles already present in the literature on the subject and provide a hypothesis to support the purposes of the study.

Author Response

POINT-BY-POINT RESPONSES TO THE REVIEWERS

Dear editors,

We would like to thank you and the reviewers for the editorial management, comments and suggestions on the paper. Below please find our point-by-point response to the reviewers.

Sincerely

Alberto Zambrano

Reviewers’ comments:

Reviewer #1 (Comments for the Author):

I think that the data presented is interesting, however I am not entirely sure of the correctness of the model. Although the bleomycin model is commonly used in mice because it can reproduce many aspects of IPF, administration of bleomycin is more suitable for inducing acute lung injury. For this reason I think that bleomycin treatment may be useful in inducing DNA damage but cannot be considered an absolute model of IPF. I believe that this observation should at least be reflected in the limitations of the study, which are not present.

OUR RESPONSE:

The observation suggested has been included in the text (lines: 50-54)

Although bleomycin reproduces well many aspects of the general pulmonary fibrosis and some lesions present in IPF has never been promoted as an experimental equivalent of IPF. The strength of the bleomycin model consists of the reproducibility and versatility as a model of general fibrosis. In addition, its high efficiency levels inducing DSBs makes bleomycin a very interesting model to analyze DNA damage (17).

Furthermore, the discussion is extremely heated. I advise the authors to cite and comment on other articles already present in the literature on the subject and provide a hypothesis to support the purposes of the study.

OUR RESPONSE:

The discussion has been extended, We have explained the limitations of the bleomycin models and its advantage in vitro for the study of DNA damage, and added references (text lines 51-55 and from 460-468):

Lines 51-55:

“Although bleomycin reproduces well many aspects of the general pulmonary fibrosis and some lesions present in IPF has never been promoted as an experimental equivalent of IPF. The strength of the bleomycin model consists of the reproducibility and versatility as a model of general fibrosis. In addition, its high efficiency levels inducing DSBs makes bleomycin a very interesting model to analyze DNA damage (17).”

Lines 460-465:

“Bleomycin can induce pulmonary fibrosis and fibrogenic cytokine release by oxidant-mediated DNA scission in a variety of animal models. The principal drawbacks of the bleomycin model relate to the rapid lung remodeling and the emphysema-like changes induced (17). However, it reproduces well many aspects of the general pulmonary fibrosis and some lesions present in IPF although it should be stated here that bleomycin is not a reliable model of IPF (17). However, the potential of bleomycin in the induction of DSBs and senescence in many cell types is extraordinary (13) (10). We have make use of this advantage in the present study to explore the influence of various vitamin D hypocalcemic analogs in the context of A549 cells and minilungs generated from hESCs.”

The working hypothesis was stated before in text lines (64-66) and now also in the discussion part (text lines 483-485):

“As a continuation of our earlier work (10), our current hypothesis states that hypocalcemic vitamin D analogs could show a lower incidence in the expression of DNA damage upon a bleomycin insult than the active form of vitamin D.”

Reviewer 2 Report

Hello,

The manuscript titled “ Effects of hypocalcemic vitamin D analogs in the expression of DNA damage induced in minilungs from hESCs: implications for lung fibrosis” needs some minor and major correction before approval.

Minor:

  1. Some typo/space error in manuscript.
  2. The graphs on page 14 Figure B and D are not align to the paper size. Please re adjust for better view.
  3. The sample size should be expanded, limiting to n=3 does not justify the significance of cellular work.

Minor:

  1. The discussion part requires more arguments and justification of the present study and results.
  2. The limitation and future direction aspects needs be added.

Author Response

POINT-BY-POINT RESPONSES TO THE REVIEWERS

Dear editors,

We would like to thank you and the reviewers for the editorial management, comments and suggestions on the paper. Below please find our point-by-point response to the reviewers.

Sincerely

Alberto Zambrano

Reviewers’ comments:

Reviewer #2 (Comments for the Author):

Hello,

The manuscript titled “ Effects of hypocalcemic vitamin D analogs in the expression of DNA damage induced in minilungs from hESCs: implications for lung fibrosis” needs some minor and major correction before approval.

Minor:

  1. Some typo/space error in manuscript.
  2. The graphs on page 14 Figure B and D are not align to the paper size. Please re adjust for better view.
  3. The sample size should be expanded, limiting to n=3 does not justify the significance of cellular work.

OUR RESPONSE:

We have revised the text and corrected typos found. We have readjusted the size of Figure 4 present in page 14. Along the indicated “n”, the number of cells analyzed or the number of organoids employed in each replicate are indicated. We consider that extending the sample size could be done in the next work focused on the in vivo implications of the present work. 

Regarding the adjustment of the figure 4, The paper size in illustrator is DIN A4; pdf viewer show it also well. Perhaps, the was a problem with margins during the elaboration of the draft. I will fix this issue with the editors. Anyway, I can provide to the system the individual figure 4.

Minor:

  1. The discussion part requires more arguments and justification of the present study and results.
  2. The limitation and future direction aspects needs be added.

OUR RESPONSE:

The discussion has been extended, We have explained the limitations of the bleomycin models and its advantage in vitro for the study of DNA damage, and added references (text lines 51-55 and from 460-468):

Lines 51-55:

“Although bleomycin reproduces well many aspects of the general pulmonary fibrosis and some lesions present in IPF has never been promoted as an experimental equivalent of IPF. The strength of the bleomycin model consists of the reproducibility and versatility as a model of general fibrosis. In addition, its high efficiency levels inducing DSBs makes bleomycin a very interesting model to analyze DNA damage (17).”

Lines 460-465:

“Bleomycin can induce pulmonary fibrosis and fibrogenic cytokine release by oxidant-mediated DNA scission in a variety of animal models. The principal drawbacks of the bleomycin model relate to the rapid lung remodeling and the emphysema-like changes induced (17). However, it reproduces well many aspects of the general pulmonary fibrosis and some lesions present in IPF although it should be stated here that bleomycin is not a reliable model of IPF (17). However, the potential of bleomycin in the induction of DSBs and senescence in many cell types is extraordinary (13) (10). We have make use of this advantage in the present study to explore the influence of various vitamin D hypocalcemic analogs in the context of A549 cells and minilungs generated from hESCs.”

The working hypothesis was stated before in text lines (64-66) and now also in the discussion part (text lines 483-485):

“As a continuation of our earlier work (10), our current hypothesis states that hypocalcemic vitamin D analogs could show a lower incidence in the expression of DNA damage upon a bleomycin insult than the active form of vitamin D.”

Future direction in terms of in vivo work has been included in the last sentence of the discussion

Round 2

Reviewer 2 Report

Hello Authors,

Thank you for submitting your work after addressing the comments. There are some areas which requires further attention.

Minor:

  1. The discussion part is an opportunity to elaborate the study aim. Analyzing the current published work and pattern and then using it to justify your work strengthen the hypothesis of the study. Simply discussing your results does not make strong point for the scientific readers to understand the impact of your study. Please take a time to revise the discussion part before next submission.
  2. There are still many areas that needs attention for figure clarity. Page 13 figure 4C and 4G, the caption has some missing part which may be omitted while copy paste process.
  3. Still the figure 4B and 4D are not seen proper. Hopefully it can be fixed before publication with editors help.
  4. Please re look the font style and size for all the figure, figure legends and titles.

Author Response

RESPONSE TO THE REVIEWER

Reviewers’ comments:

Reviewer #2 (Comments for the Author):

Hello Authors,

Thank you for submitting your work after addressing the comments. There are some areas which requires further attention.

Minor:

  1. The discussion part is an opportunity to elaborate the study aim. Analyzing the current published work and pattern and then using it to justify your work strengthen the hypothesis of the study. Simply discussing your results does not make strong point for the scientific readers to understand the impact of your study. Please take a time to revise the discussion part before next submission.

OUR RESPONSE:

We have revised the Discussion section accordingly, elaborating better the study aim and analyzing published reports.

Discussion:

Besides its function in the mineral homeostasis and immune system, vitamin D plays a role in multiple chronic diseases involving the respiratory system. Epidemiological studies have suggested a link between vitamin D deficiency and the risk of development of chronic lung diseases such as asthma, chronic obstructive pulmonary disease (COPD), cystic fibrosis and respiratory infections (31). This association has led to the notion that vitamin D supplementation might ameliorate the progress of these diseases. Vitamin D supplementation, however, needs to be evaluated carefully as it can be a factor contributing to vitamin D-mediated hypercalcemia and hypercalciuria (32). In addition, it has been reported cases of vitamin D toxicity associated to overdoses due to manufacturing or intake errors (33). Moreover, we have reported a detrimental role of vitamin D supplementation, in a therapeutic experimental system, very likely associated to an impairment of the cellular DSBs repair capabilities and cell senescence (10).  

Vitamin D may affect the progression of fibrosis at different stages: anti-fibrinolytic coagulation cascade, inflammation, fibroblasts activation and the negative regulation of the renin-angiotensin system. Vitamin D seemed to prevent the experimental lung fibrosis induced by bleomycin (8) (9) (34) (35) (36). However, in these experimental studies, vitamin D is administered either before or very early after the bleomycin insult so the effects observed were very likely due to the inherent anti-inflammatory properties of vitamin D. Thus, these studies can be defined as preventive. In addition, various hypocalcemic analogs such as paricalcitol, calcipotriol and 22-oxacalcitriol, have been proved to be active as anti-fibrotic agents in different experimental systems and types of fibrosis (37) (38) (39) (40) (41) (42) (43) (44) (45)  (46) (47) (48) (49). Vitamin D less-hypercalcemic analogs might provide an alternative to vitamin D supplementation to treat many conditions related to fibrosis. The ideal analog would retain vitamin D receptor binding capacities and have minimal effects on mineral metabolism.    

Bleomycin can induce pulmonary fibrosis and fibrogenic cytokine release by oxidant-mediated DNA scission in a variety of animal models. The principal drawbacks of the bleomycin model relate to the rapid lung remodeling and the emphysema-like changes induced (17). However, it reproduces well many aspects of the general pulmonary fibrosis and some lesions present in IPF although it should be stated here that bleomycin is not a reliable model of IPF (17). However, the potential of bleomycin in the induction of DSBs and senescence in many cell types is extraordinary (13) (10). We have make use of this advantage in the present study to explore the influence of various vitamin D hypocalcemic analogs in the context of A549 cells and minilungs generated from hESCs. Our experimental approaches the initial steps of the fibrogenic conditions, i.e., the expression of DNA damage underlying many conditions evolving towards fibrosis.

The generation of human minilungs which share the structural features and some extent of the functionality of the native organ may serve as system model to emulate the DNA damage inflicted during the course of fibrogenic conditions such as IPF. Currently, the more efficient protocols to generate airway and alveolar epithelial cells from the direct differentiation of hPSCs are biased to the production of alveolar cells (22) (27) (29). We have employed either shocks or continuous exposures of bleomycin. The sublethal bleomycin shocks employed here allow the accurate quantification of DNA damage in the form of DSBs and the observation of subtle differences between the experimental conditions that might be otherwise masked by the extraordinary potential of bleomycin. Bleomycin seems to inflict DNA damage in the form of DSBs in all the epithelial cells equally, even when the cell organization is the form of lung buds embedded in MatrigelTM sandwiches. However, the assembly of organoids into MatrigelTM sandwiches can make difficult the access of bleomycin and ligands to the cells. The reduction in the extent of DNA damage inflicted by bleomycin compared to the 2D minilungs or A549 cells might reflect this fact.

As a continuation of our earlier work (10), our current hypothesis states that hypocalcemic vitamin D analogs could show a lower incidence in the expression of DNA damage upon a bleomycin insult than the active form of vitamin D. The initial results presented here suggest that less-hypercalcemic analogs do not show the deleterious effects observed by vitamin D treatment in the presence of bleomycin and could be an alternative to vitamin D supplementation. In addition, the treatment with this kind of vitamin D analogs could be tested as efficient agents to reduce the bulk of DD expression underlying multiple diseases that can evolve with DNA damage, fibrosis and aging such as IPF and other lung interstitial conditions. Future in vivo work in this direction will be necessary. 

  1. There are still many areas that needs attention for figure clarity. Page 13 figure 4C and 4G, the caption has some missing part which may be omitted while copy paste process.
  2. Still the figure 4B and 4D are not seen proper. Hopefully it can be fixed before publication with editors help.

OUR RESPONSE:

I do not know the reason of this problem. We are uploading Figure 4 in PDF and EPS formats for the reviewer and the editors. I feel the problem could be related to inclusion of the Figure.   

  1. Please re look the font style and size for all the figure, figure legends and titles.

OUR RESPONSE:

We have modified font styles and sizes to homogenize all the figures as suggested. We are providing the new figures in PDF format, and figure 4 in EPS format too. We have revised the figure legends and the titles. The current edition uses the format of the journal.